# TAVI-PREP: A Deep Learning-Based Tool for Automated Measurements Extraction in TAVI Planning

**DOI:** 10.3390/diagnostics13203181

**Published:** 2023-10-11

**Authors:** Marcel Santaló-Corcoy, Denis Corbin, Olivier Tastet, Frédéric Lesage, Thomas Modine, Anita Asgar, Walid Ben Ali

**Affiliations:** 1Montreal Heart Institute, Montreal, QC H1T 1C8, Canada; 2Faculty of Medicine, University of Montreal, Montreal, QC H3T 1J4, Canada; 3Department of Electrical Engineering, Polytechnique Montreal, Montreal, QC H3T 1J4, Canada; 4Hôpital Haut Lévêque Bordeaux, 33600 Pessac, France

**Keywords:** transcatheter aortic valve implantation (TAVI), deep neural networks, automatic preoperative planning

## Abstract

Background: Transcatheter aortic valve implantation (TAVI) is a less invasive alternative to open-heart surgery for treating severe aortic stenosis. Despite its benefits, the risk of procedural complications necessitates careful preoperative planning. Methods: This study proposes a fully automated deep learning-based method, TAVI-PREP, for pre-TAVI planning, focusing on measurements extracted from computed tomography (CT) scans. The algorithm was trained on the public MM-WHS dataset and a small subset of private data. It uses MeshDeformNet for 3D surface mesh generation and a 3D Residual U-Net for landmark detection. TAVI-PREP is designed to extract 22 different measurements from the aortic valvular complex. A total of 200 CT-scans were analyzed, and automatic measurements were compared to the ones made manually by an expert cardiologist. A second cardiologist analyzed 115 scans to evaluate inter-operator variability. Results: High Pearson correlation coefficients between the expert and the algorithm were obtained for most parameters (0.90–0.97), except for left and right coronary height (0.8 and 0.72, respectively). Similarly, the mean absolute relative error was within 5% for most measurements, except for left and right coronary height (11.6% and 16.5%, respectively). A greater consensus was observed among experts than when compared to the automatic approach, with TAVI-PREP showing no discernable bias towards either the lower or higher ends of the measurement spectrum. Conclusions: TAVI-PREP provides reliable and time-efficient measurements of the aortic valvular complex that could aid clinicians in the preprocedural planning of TAVI procedures.

## 1. Introduction

Transcatheter aortic valve implantation (TAVI) has emerged as a less invasive alternative to open-heart surgery for treating severe aortic stenosis. The number of patients eligible for TAVI is growing fast due to the aging population, as are robust results of this technique in patients at intermediate [1,2,3] and low [4,5] surgical risk. This has led to current guidelines suggesting TAVI to a broader array of patients, including those at low surgical risk [6,7].

Preoperative procedures for TAVI include gathering specific anatomical measurements of the aortic valve complex to guide the choice of instruments and manufactured prosthetic valve. Measures of the minimum annulus area diameter ensure that there is enough space for the transcatheter to deploy the manufactured valve. The aortic annulus perimeter and area help to guide the decision for the valve type used and, consequently, its size. Paravalvular leaks could result from underestimating the aortic annulus, while an overestimation could lead to annulus rupture and other complications, which could necessitate pacemaker implantation [8,9]. Coronary obstruction is among the most severe potential complications post-TAVI. It is associated with low coronary ostia height and a narrow sinus of Valsalva. Patients presenting coronary obstruction following the TAVI procedure (between 0.4% and 1.2% of cases) have a mortality rate of 50% [10]. Coronary heights can therefore distinguish patients that might be at risk for this procedure, since the coronaries could be obstructed by the prosthetic valve. More specifically, coronary heights smaller than 10 mm, especially when the sinus of Valsalva is narrow (less than 28 mm), have higher coronary occlusion risk [11]. Obtaining accurate anatomical measurements is therefore crucial to make appropriate decisions regarding the valve type and size and to predict complications that would require careful follow-up or that might make the procedure unsuited for specific patients.

Computed tomography (CT) angiography is the preferred imaging modality used to produce essential accurate preoperative assessments, and a consensus has been published on the correct technique for obtaining and analyzing these scans [12]. However, with existing planning software, measurements are performed semi-automatically [9] and are time-consuming and prone to variability [13,14] compared to fully automatic solutions. Hence, the full automation of the entire process, including segmentation, landmark detection, and measurement extraction, is still lacking.

The use of machine learning techniques has revolutionized patient care by strengthening four domains: the processing and interpretation of digital images, predictive modelling, precision-medicine-guided recommendations, and the ability to act as health system performance enhancers [15]. Among them, the field of medical diagnosis is one where this tool is best suited, as deep learning algorithms and convolution neural networks can work efficiently on a narrow task in a predefined context to upgrade the process of visual pattern recognition [16]. These techniques have been estimated to be 5% to 10% more accurate than the average clinician [17]. However, the performance of the algorithms is greatly dependant on the datasets used to frame them, so ensuring the good quality of the data is crucial.

In the specifics of TAVI procedures and outcomes, numerous deep learning applications have been reported in the literature in recent years [18]. Among them, some works have attempted to automatize the measurements of some of the anatomical structures. Krüger et al. [19] were able to obtain the area-derived diameter within the aortic annulus region with a mean error below 2 mm between the automatic measurement and the diameter derived from annotations on 36 subjects. Similarly, Ma et al. [20] reported a mean error of 2.23 mm on 150 individual CT volumes against manual annotations. Saitta et al. [21] successfully obtained measurements of the aortic annulus, sinotubular junction, and sinuses. They found small discrepancies (5%) between the manually measured values and the automatic measurements for area-related parameters, while larger discrepancies (10%) were observed for perimeter-related measurements. Astudillo et al. [22] aimed to detect landmarks, specifically the nadirs and coronary ostia positions. They achieved an average error of 1.5 mm when comparing the model’s predictions to the ground truth annotations in a sample of 100 patients.

However, challenges remain in the field of pre-TAVI measurements, and we aim to address them comprehensively through our work. These challenges include the absence of a fully automated algorithm, limitations in validation cohorts, dependence on a single expert’s input, and the omission of important measurements. In response to these issues, our work can be summarized as two key contributions:Full Automation: We introduce a fully automated pre-TAVI measurement extraction algorithm capable of extracting 22 measurements in approximately 2 min, streamlining clinical workflows.Robust Validation: Our algorithm undergoes comparison with two experts in the field, enhancing its reliability and suitability for potential future clinical use. Validation is conducted on the largest cohort to date, involving 200 patients, further ensuring its accuracy and applicability.

In this work, we propose a fully automatic pipeline based on a deep learning method for the automatic extraction of pre-TAVI measurements from CT scans using a mesh generation technique and landmark identification. These measurements are then compared to a manual evaluation made by experienced cardiologists. The ability of the algorithm to deal with edge cases and potential sources of error is detailed. Finally, its accuracy is compared to other works reported in the literature. The combination of these techniques has the potential to improve the workflow of pre-TAVI planning by allowing the accurate identification of anatomical structures in a more time-efficient way.

## 2. Materials and Methods

This study proposes a comprehensive, fully automated preoperative planning framework for TAVI. The framework integrates the segmentation technique presented by Kong et al. [23], alongside landmark detection methods inspired by Astudillo et al. [22]. The segmentation method leverages deep learning algorithms to generate accurate and detailed anatomical segmentations from routine pre-TAVI computed tomography (CT) scans. Deep-learning techniques are also used to identify key anatomical features and landmarks in imaging data. The proposed method requires approximately 2 min to predict all 22 measurements, an improvement compared to the expert’s average time of 15 min for the same task. The complete workflow can be visualized in Figure 1.

### 2.1. Data

#### 2.1.1. Segmentation Dataset

For the segmentation data, the public MM-WHS dataset [18,19,20] was used. Training was performed on the publicly available CT scans and 15 in-house manual segmentations provided by an expert cardiologist. A total of 35 images were used for training the algorithm. Training regimens were the same as those initially published by Kong et al. [23].

#### 2.1.2. Landmark Detection Dataset

A total of 104 individuals were selected for this dataset. Male subjects, who composed 48% of the cohort, had an average age of 78.9 ± 6.2, and female subjects’ average age was 80.8 ± 5.9. They were all candidates for TAVI and exhibited some level of aortic valve calcification. An ECG-gated contrast CT angiography of the aortic root and heart was performed on these patients as part of their TAVI preoperative planning using a Siemens SOMATOM Force CT machine, located at the Montreal Heart Institute (MHI). The resultant images were of the dimension (512 × 512 × *n*), with *n* denoting a slice thickness of 1 mm. Images had pixel spacing ranging from 0.256 to 0.525 mm for X and Y coordinates, respectively. The expert cardiologist annotated landmarks on all 104 scans for training and validation (70, 17, 17). Images were resampled with a pixel spacing of 1 × 1 × 1 mm, preprocessed with CLAHE [24] and then normalized between 0 and 1.

#### 2.1.3. Final Measurements Dataset

For the final measurement dataset, 200 patients were randomly selected for predictive performance assessment (no overlap between these patients and the ones in the training and validation sets of the landmark detection dataset). The average age of this dataset (*n* = 200) was 79.8 ± 6.4 and 79.2 ± 6.9 for males and females, respectively, and 55% of the cohort were men. The cohort was representative of the Canadian national average for Transcatheter Aortic Valve Implantation (TAVI) procedure, which reports an average age of 81.6 ± 7.6 with 54.7% of patients being men [25]. Image spacing and slice thickness were the same as the landmarks’ detection dataset. 3Mensio Structural Heart version 10.3 (3Mensio Medical Imaging, Bilthoven, The Netherlands) [26], a widely accepted gold standard software for semi-automated pre-TAVI planning, was employed for the expert cardiologist’s measurements. Some of the manual measurements (*n* = 115) were performed by a second expert cardiologist to evaluate the inter-observer variability.

### 2.2. Segmentation

In this study, MeshDeformNet [23] was used and customized to generate high-definition 3D surface meshes from pre-TAVI CT scans. The deep learning model was trained on the public MM-WHS (*n* = 20) dataset and an in-house private dataset (*n* = 15), focusing only on the aortic root and left ventricle volumes. Images from the public dataset underwent cropping to conform to the imaging field of view employed at MHI, which was smaller than the one used in the public dataset due to concerns related to radiation exposure. The modification did not hinder the algorithm’s generalization capability, as the field of view could be automatically reduced post-acquisition, with the region of interest centered on the left ventricle and aortic root. Other preprocessing steps, training parameters and hyperparameter tuning procedures were the same as those presented in the original article by Kong et al. [23] MeshDeformNet uses 3D convolution operations in the image encoding module to reduce the size and dimensionality of the images, which are later reconstructed in the segmentation module as binary masks associated with each structure of interest. In the encoding process, features are extracted at each stage. These features form the latent space which offers a rich informational representation of the input scan. Each filter in a convolutional layer captures different aspects of the input data, such as contours, textures, or more complex patterns. As the data pass through multiple convolutional layers, the network can learn to extract increasingly higher-level features, representing more abstract concepts or structures present in the input. These features are then used by the mesh deformation module to guide the graph convolution layers that deform the mesh. From the features of the images and resulting meshes, the final prediction of the mesh can be obtained. Expert segmentations are used as a supervisory tool, allowing the network to reach a state where the features of the latent layers of each convolution are representative of the task, namely segmentation, which guides the deformation of meshes.

Working with a mesh rather than with pixels allows for more precise measurements, first because the boundaries of the mesh prevent the risk of outlier values by being a closed surface. This also provides better resistance to noise in the images, which is prevalent in this cohort as all individuals have severely calcified aortic valves. Second, the number of points that can be assigned to the final segmentation no longer depends on the initial pixel density. As a result, compared to voxel-based segmentation, meshes can provide smoother and more natural-looking surfaces which are easier to measure. Finally, the mesh representation of the segmentation is continuous, making it more robust to small artifacts that could cause sharp edges in a traditional voxel-based segmentation.

### 2.3. Landmark Detection

The accurate segmentation produced by our implementation of MeshDeformNet can be used to identify clinically relevant landmarks. The same reference points used by experts to derive measurement are needed in a fully automatic approach. Consequently, we integrated a 3D Residual U-Net [27] supplemented by Squeeze and Excitation layers [28] specifically for landmark detection, such as coronary ostia and the three nadir positions. This methodology was inspired by previously published studies, which demonstrated notable success in deploying deep learning models for similar tasks [22].

Manually annotated landmarks were transformed into spherical binary masks with a radius of 5. Following hyperparameter tuning, the 3D Residual U-Net was trained with a Dice Loss, an SGD optimizer, a learning rate of 0.002, a batch size of 8, and with 16 feature maps embedded within the model’s architecture. The mentioned hyperparameters were fine-tuned using WandB’s (https://www.wandb.com/ accessed on 31 August 2023) Bayesian sweep functionality over a series of one hundred sweeps. This method automates the search for the best combination of hyperparameters by iteratively running experiments with different settings. This approach employs Bayesian optimization, utilizing a probabilistic model to predict which hyperparameter configurations are likely to perform best based on previous results. It efficiently explores the hyperparameter space, leading to improved model performance while saving time and resources compared to manual tuning. Results are logged and visualized, ultimately yielding an optimized hyperparameter configuration for the final model.

The number of output channels was fixed at five, corresponding to each anatomical landmark. The output segmentation masks were binarized, utilizing a threshold value of 0.5, and only the largest connected element was retained as the final binary mask. Following this, we calculated the mask’s centroid to derive the voxel coordinates of the landmark. Finally, these voxel coordinates were translated back into the original LPS coordinate system used in the initial images.

### 2.4. Derivation of Measurements

In the automation pipeline’s last phase, complex measurements, often challenging and time-consuming even for an expert cardiologist, are derived using mesh representations and anatomical landmarks. The first step of this algorithm is to define the centerline passing through the center of the lumen of the aortic root and the left ventricle. This centerline will then be used to slide planes across the meshes for measurement extraction. Figure 2 graphically represents each measurement.

#### 2.4.1. Centerline Extraction

Centerline extraction was performed with Laplacian Based Contraction [29], which uses a combination of local Delaunay triangulation and topological thinning. Point clouds are contracted and maintain the global shape of the input model by anchoring points chosen by an implicit Laplacian smoothing process. Meyer et al. [30]’s implementation was used.

#### 2.4.2. Annulus Plane

The annulus plane is defined by the 3 nadirs (non-coronary nadir (NC), right coronary nadir (RC), and left coronary nadir (LC)). A plane equation is derived from the 3 points in 3D space previously identified by the landmark detection step. The plane is then intersected with the left ventricle and aortic mesh to obtain a cross-section. This cross-section is then used to compute the area, perimeter, minimum and maximum diameters, and derived measures (area derived diameter and perimeter derived diameter). The minimum and maximum diameters are diameters that pass by the centroid of the intersection surface.

#### 2.4.3. Left Coronary Height (LCH) and Right Coronary Height (RCH)

Left and right coronary heights are obtained from the left and right coronary ostia obtained from the landmark detection algorithm. These points can be projected into the annulus plane to compute their perpendicular height.

#### 2.4.4. Left Ventricular Outflow Track (LVOT)

Based on the cardiologist’s expertise, the LVOT can be approximated as a plane located 4 mm under the aortic annulus plane. Therefore, the aortic annulus plane will be shifted downwards by 4 mm along the previously defined centerline. The displaced plane will then be intersected with the left ventricle mesh to obtain the intersection surface. Then, the same measurements can be derived (area, perimeter, minimum and maximum diameters).

#### 2.4.5. Sinus of Valsalva (SOV) and Sinotubular Junction (SNTJ)

The SOV (sinus of Valsalva) is determined as the plane where the aortic root reaches its maximum size, while the SNTJ (Sinotubular Junction) plane is defined as the point where the aorta and aortic root meet, indicating the location where the cross-sectional area of the anatomy becomes stable. To locate these planes, a list of evenly spaced planes (1 mm apart) is established along the centerline within the aortic root. For each plane along the centerline, the cross-sectional area is calculated. The objective is to enable the profiling of the cross-section area throughout the entire aortic root. The plane with the greatest area is identified as the SOV, while the plane where the area starts to stabilize is recognized as the SNTJ. For the SNTJ, the minimum and maximum diameters are computed. For the SOV plane, previously identified nadirs (NC, RC, and LC) are projected onto the plane. Lines passing through the projected points on the plane and the centroid of the plane are defined for each sinus. The diameter of each sinus is calculated by measuring the distance between the two intersection points of each respective line on the perimeter of the SOV plane’s cross-sectional surface.

## 3. Results

### 3.1. Segmentation and Landmark Detection Performance

The segmentation performance of the algorithm used in this paper has already been discussed in detail in Kong et al. The algorithm performed as well qualitatively on the images of the MM-WHS test set as on the images acquired at the MHI. Since all available data with an associated ground truth mask were used to train the algorithm and manual segmentation is very time consuming for the expert, this part of the algorithm was not quantitively validated at this step. The landmark detection performance was evaluated on a test set (*n* = 17). An average MAE of 2.01 ± 0.5 mm was reported.

### 3.2. Manual vs. Automatic Measurements

The end goal was to achieve automatic measurements as accurate as those provided by an expert cardiologist, or at least as good as the expected inter-individual variability between experts. Thus, the performance of the algorithm was evaluated in the following ways:
Mean relative error, correlation coefficients, and confidence intervals (CIs): Discrepancies were reported as the absolute relative mean of the error and the 95% confidence interval (*CI*) boundaries, as defined in Equation (1). $\tilde{x}$ is the mean, *Z* is the chosen z-score (1.96 for 95% *CI*), *s* is the standard deviation, and *n* is the number of samples. Pearson correlation coefficients are also reported.
(1)$$CI=\tilde{x}\pm Z\frac{s}{\sqrt{n-1}}$$Bland–Altman plots: A graphical method to analyze the agreement between two quantitative measurements. Plots were created in a pairwise fashion (Expert 1 vs. Expert 2, TAVI-PREP vs. Expert 1, and TAVI-PREP vs. Expert 2). These plots give a comprehensive understanding of how predicted values compare to expected values across the range of measurements.


### 3.3. Confidence Intervals and Pearson Correlation Coefficients

#### 3.3.1. Annulus and LVOT

The absolute mean relative errors and correlation coefficients fell within the range of measurements made by experts. Concerning relative errors, the mean percentage of differences between experts was slightly smaller than the percentage of error between automated measurements and both expert reports (Figure 3a). This pattern was mirrored in the correlation coefficients, where the correlation coefficients between experts were slightly higher than when, respectively, they were compared to automated measures (Figure 3b). Nonetheless, the confidence intervals of the automated measurements against both experts, respectively, overlapped with the confidence intervals of the expected variability between expert measurements, implying that the predictions fell within an acceptable margin of error.

#### 3.3.2. SNTJ

The percentage of error between the automated measurements and those reported by expert1 was smaller than the expected difference between experts (Figure 3a). Consequently, correlation coefficients were higher between the algorithm and Expert 1.

#### 3.3.3. Sinus

The predicted measurements for left coronary commissure (LCC), right coronary commissure (RCC), and non-coronary commissure (NCC) sinus diameters diverged from the measurements provided by the two experts by an average of 0.72%. More precisely, the differences between experts were 2.84% [2.42, 3.25], 3.05% [2.67, 3.41], and 1.95% [1.65, 2.25] for LCC, RCC, and NCC, respectively. Compared to Expert 1, TAVI-PREP had relative errors of 2.82% [2.42, 3.23], 4.08% [3.57, 4.59], and 3.05% [2.64, 3.45] for LCC, RCC, and NCC, respectively. Realistically speaking, this converts to a 95% CI with an average variability of 5.02 mm (95% CI: [−2.32, 2.70]) between TAVI-PREP and Expert 1 while the variability is of 3.44 mm (95% CI: [−1.14, 2.30]) between the two experts for the three values.

#### 3.3.4. Coronary Heights (LCH and RCH)

Predicting the right coronary height proved to be the most challenging, with a mean absolute error of 2.82 mm [−2.06, 7.71] compared to Expert 1. LCH predictions were more in agreement with the experts’ predictions, with a mean absolute error of −0.05 mm [−4.00, 3.89] Furthermore, the distribution of errors for both the left coronary height and right coronary height was wider than that of the experts (Figure 4c vs. Figure 4d for RCH). As a result, the Pearson correlation coefficients were low, with values of 0.80 [0.74, 0.85] for the left coronary height and 0.72 [0.64, 0.78] for the right coronary height compared to experts, who had values of 0.92 mm [0.89, 0.94] and 0.86 [0.82, 0.90], respectively.

Overall, the analysis highlights that while the automatic predictions generally aligned with the experts’ measurements, there were specific areas of both agreement and discrepancy. The predictions tended to exhibit a closer agreement with one expert (Expert 1) in some cases, while in others there were challenges in predicting specific measurements accurately (sinus diameters and coronary heights). Correlations, 95% Cis, mean absolute errors, and absolute mean relative errors for all measurements are available in Appendix A.

### 3.4. Bland–Altman

Bland–Altman plots were also used to assess the degree of consistency between the automatic measurements and the manual measurements carried out in 3Mensio by the expert cardiologist. These plots can be useful in visualizing the consistency of the predictions with experts’ measurements. Bland–Altman plots were generated as follows: Expert 1–TAVI-PREP, Expert 1–Expert 2 and Expert 2–TAVI-PREP. All Bland–Altman plots are available in Appendix A.

The differences in measurements (Figure 4) demonstrate a higher level of consensus among experts than when compared to the automatic approach, as evidenced with narrower confidence intervals around the mean differences for the corresponding measurements (95% CIs of [−60.65, 41.36] for Figure 4a compared to [−49.30, 33.14] for Figure 4b and [−2.06, 7.71] compared to [−2.35, 4.44]). The density plots do not exhibit a discernible bias towards either the lower or higher ends of the measurement spectrum. Notably, Case 106 is an outlier in our algorithm’s predictions. This specific case also poses a clinical challenge since the experts do not agree on the measures either (Figure 4). It also means that the algorithm agrees with Expert 2 in this case (See Appendix A).

### 3.5. Edge Cases

Edge cases are infrequent due to the algorithm’s inherent redundancy and ability to automatically identify poorly predicted measurements (such as the incorrect identification of nadirs, SNTJ, or SOV planes), with potential correction mechanisms (like a secondary estimation of nadirs position). However, there were still instances labeled as outliers, primarily attributed to segmentation challenges. Some scans exhibited artifacts of varying degrees, which could significantly impact the segmentation process and lead to less-than-optimal performance. Moreover, conspicuous calcification can detrimentally affect the accuracy of segmentation, subsequently affecting the final measurements. In Figure 5, an example is presented where aortic root segmentation failed to accurately capture the anatomy due to complexities posed by coronary ostia, resulting in unexpected segmentation irregularities.

Figure 6 presents case 106, an instance where the left ventricle segmentation was inaccurate due to the presence of a sizable calcification plaque. These errors rippled through subsequent measurements, as depicted in the figures. Case 106 is a challenging case due to the high level of calcification present in the anatomical regions of interest (LVOT and aortic annulus).

## 4. Discussion

Measurements in 3Mensio were carried out in a semi-automatic manner where the nadirs’ positions (used to define the aortic annulus plane) were suggested by the software, then corrected by the user if needed. The LVOT was defined as the plane 4 mm under the annulus plane along the centerline. Thus, all measurements related to the LVOT and aortic annulus were tightly linked together, meaning that an error in the initial nadirs’ location identification would also affect LVOT measurements. As shown in Figure 3, TAVI-PREP still managed to achieve similar correlation coefficients and mean absolute relative errors that were within the bounds of the confidence intervals of the experts. SNTJ correlations were lower between experts compared to annulus and LVOT. This could be explained by the fact that the SNTJ plane was selected as the plane where the cross-sectional area started to decrease as you moved along the centerline of the aortic root. The plane area was visually approximated by the clinician carrying out the analysis on 3Mensio, making it subject to higher inter-user variability. Similarly, the minimum and maximum SNTJ diameters were also more prone to error since the plane was less likely to be the same between users. This resulted in the algorithm being closer to Expert 1 by an average of 2% compared to Expert 2. The landmark annotation dataset was based on Expert 1’s annotations, which could explain the tendency to agree more with Expert 1 than Expert 2. The right coronary height and left coronary height are challenging measures to derive because they depend on multiple factors. First, the ostia must be correctly identified by the algorithm, which can sometimes be challenging due to uneven intensity levels in the scan due to severe calcification. Since their value is derived from the vector responsible for the projection of the ostia on the aortic annulus plane, said plane must be defined perfectly. Secondly, the nadirs must be correctly placed, and they can be challenging to identify in some case, even though there is built-in redundancy in the algorithm. Furthermore, the reported 2.01 mm error on landmark detection can greatly affect the orientation of the plane (and the ostia), giving it an unnatural tilt. Since the length of the projected vector on the plane was dependant on the plane orientation, there was an accumulating error. RCH and LCH were therefore the lackluster parts of our algorithm, which could be improved with supplemental training data, something that we are actively working on with the public dataset ImageCAS [31].

Various works in recent years have described attempts at the automatization of the measurements of some elements of the aortic complex. To our knowledge, however, TAVI-PREP is the first algorithm published to date capable of providing the whole ensemble of critical measurements of the pre-TAVI assessment, i.e., the annulus, LVOT, SOV, and SNTJ dimensions, as well as the coronary height. It also features the largest validation cohort published to date, with 200 patients. Table 1 summarizes the differences in absolute error of TAVI-PREP against previous works [21,22,32] across the different anatomical structures.

TAVI-PREP displayed a lower absolute error in the annulus area, perimeter, annulus minimum, diameter, annulus maximum diameter, annulus average diameter, and STNJ average diameter, with narrower confidence intervals. Saitta et al. reported a remarkably low absolute error of the annulus area-derived diameter, at 0.07 mm [−0.24, 0.38], which outperformed TAVI-PREP and the experts’ measurements by a large margin. However, it was one order of magnitude smaller than the errors reported on their other annulus measurements, raising the question of whether there could be a unit error in their results (should have been reported as cm instead of mm).

It is also worth citing the work of Krüger et al. [19], not displayed in the table since they provided standard deviation instead of confidence intervals, who reported a mean minimal annulus diameter of 1.84 ± 1.21, worse than TAVI-PREP. Finally, the work from Astudillo et al. outperformed the prediction capability of TAVI-PREP for coronary height. This could be in part attributed to a smaller variation in their cohort, if we consider the smaller scale on their Bland–Altman plot *x*-axis.

Accounting for the potential error in Saitta et al., this would place TAVI-PREP as the best performing algorithm out there for the whole ensemble of critical measurements, except for LCH and RCH. However, femoral access is not considered in this study, which is used for other important measurements.

The same trends can be found in Table 2, where TAVI-PREP outperformed other work in terms of correlation coefficients, except for coronary height. Nevertheless, the PCC for LCH was reported as the same as Astudillo et al., at 0.8. The average LCH-RCH confidence intervals also overlapped with the predicted value of Astudillo et al. (upper bound at 0.82 compared to the mean reported value of 0.8 by Astudillo et al.).

Multiple limitations should be considered when interpretating the results of this study. First, inter-individual variability was assessed by two expert cardiologists from the Montreal Heart Institute who received similar training, one expert being the mentor of the other. Therefore, there was an inherent bias in the correlation between the two experts, since one supervised the other during his training. This would mean that the correlation between experts could be overestimated. Therefore, the study should have included an unrelated third expert cardiologist to better understand inter-individual variability. Le Couteaulx et al. [33] carried out a study on inter-observer reproducibility using CardiQ planning software (*n* = 52). While comparing their two most experienced cardiologists, they reported a lower PCC and larger CI for annulus diameter average (0.94 [0.89, 0.96]), annulus area-derived diameter (0.94 [0.89, 0.97]), and annulus perimeter-derived diameter (0.92 [0.86, 0.96]). However, it is not mentioned if the experts were related. Another unexplored aspect in this study is the intra-individual variability. Cardiologists make their pre-TAVI measurements with the best of their knowledge and based on the training they have received, which are two continuously evolving aspects of their practice. Le Couteaulx et al. also reported expert intra-observer agreement that was similar to our reported inter-observer agreement: annulus diameter average (0.98 [0.96, 0.99]), annulus area-derived diameter (0.98 [0.96, 0.99]), and annulus perimeter-derived diameter (0.97 [0.95, 0.98]) The algorithm we proposed will always give the same results, whether it is used by an expert or an apprentice. However, this would not be the case for 3Mensio, or for any other currently available solution which needs the user to go through a non-negligeable learning process to be proficient at it [13,14,33,34]. Lastly, inter-software variability has not been explored. Software vendors usually provide their own training to their users to ensure optimal performance. Even though 3Mensio is considered state of the art, it is not the only TAVI planning software used by cardiologists. We hypothesize that adding a comparison with another software would further reinforce the stance of TAVI-PREP as the best-performing automatic solution, even compared to semi-automatic solutions. Valve selection, possible complications, and outcomes (success/failure) were not explored due to not all cases having had their TAVI procedure at the time of writing, but they will be explored in future studies. Furthermore, some measurements, such as aortic angle and femoral access diameters, were not predicted, but their inclusion would have been essential for providing accurate valve recommendations.

## 5. Conclusions

TAVI-PREP successfully extracts the key measurements for TAVI planning from CT scans in a completely automatic and time-efficient fashion. It uses surface mesh generation and a ·3D Residual U-Net for landmark detection. The mesh provides a smoother surface for the segmentation, being more robust to artifacts. It can identify poorly placed landmarks and has correction mechanisms. When compared to the evaluation made by experienced cardiologists, it yields high Pearson correlation coefficients and low mean absolute relative errors for most parameters, while struggling with the prediction of LCH and RCH. The error in landmark identification is comparable to, if not better than, that reported in previous works. This system highlights the utility of AI in planning transcatheter procedures and could prove useful amid global TAVI growth.

## Figures and Tables

**Figure 1 diagnostics-13-03181-f001:**
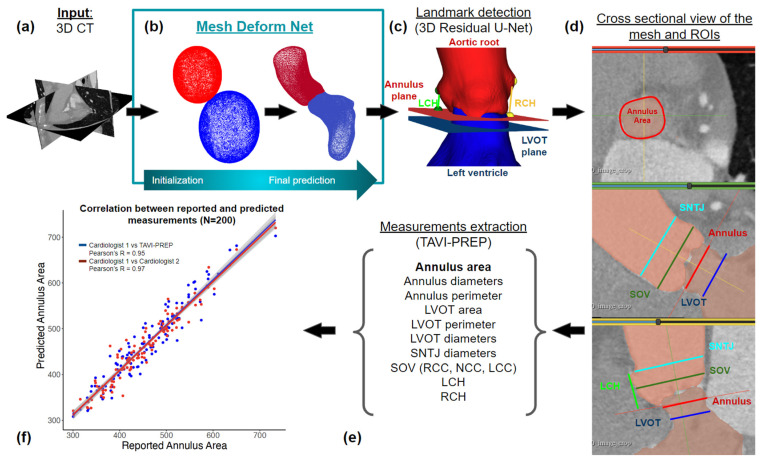
Workflow of measurement extraction from a clinical CT scan. (**a**) 3D phase specific CT scan, usually taken at 30% of the cardiac cycle. (**b**) The segmentation module, as proposed by Kong et al., allows the creation of an accurate anatomical representation of a patient’s anatomy in the form of a mesh. Spherical meshes are deformed to closely math the anatomy. (**c**) Landmark detection step, where the position of the coronary ostia and the three nadirs’ commissures is obtained. This step defines the annulus and LVOT plane position and the left and right coronary height (LCH and RCH). (**d**) The 2D cross sectional views, taken from the 3D mesh representation, used to derive measurements. (**e**) List of measurements extracted by the proposed algorithm. (**f**) Correlation plot between the expected and reported measurements for the annulus area [23].

**Figure 2 diagnostics-13-03181-f002:**
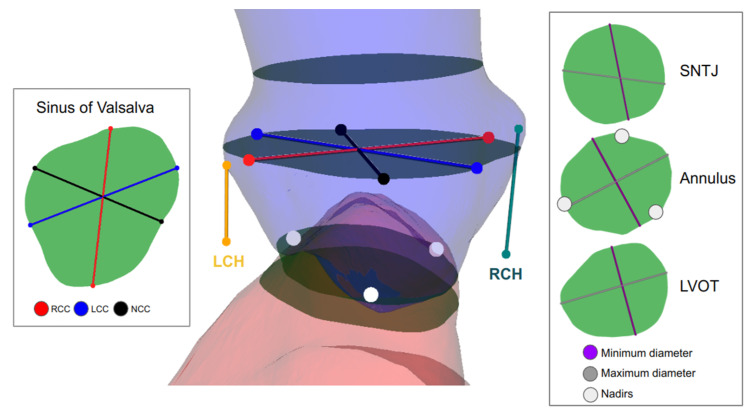
Summary of the derived measurements from our TAVI-PREP algorithm.

**Figure 3 diagnostics-13-03181-f003:**
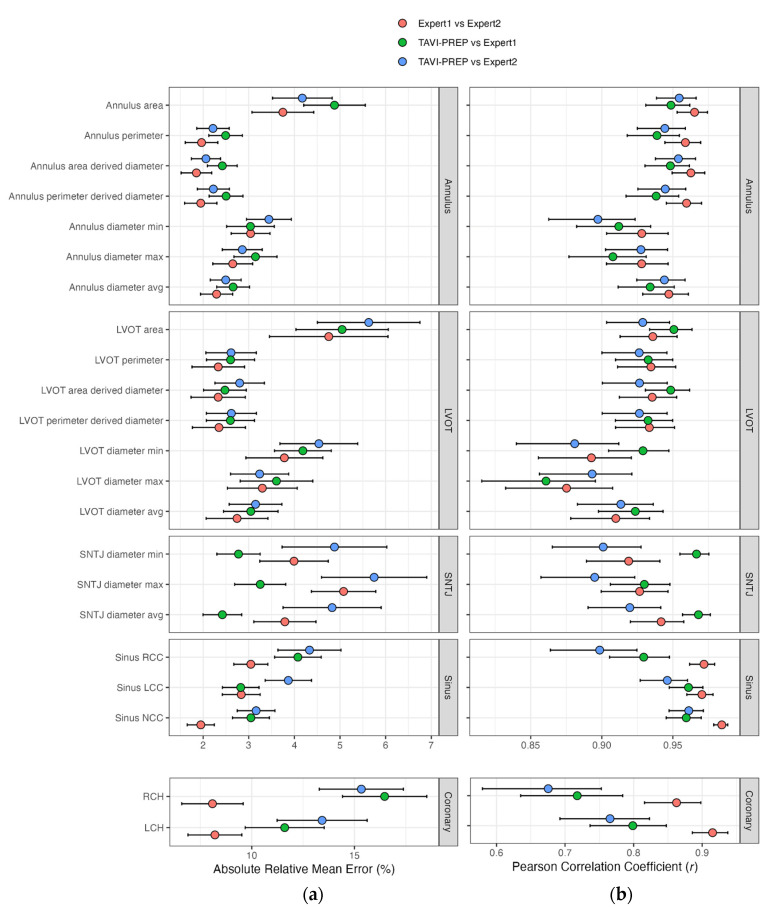
Absolute mean relative error (**a**) and Pearson correlation coefficients (**b**) with their corresponding confidence intervals for each measurement are reported. Values are reported as mean [95% CI]. The coronary scale is larger due to a lower predictive performance for these values.

**Figure 4 diagnostics-13-03181-f004:**
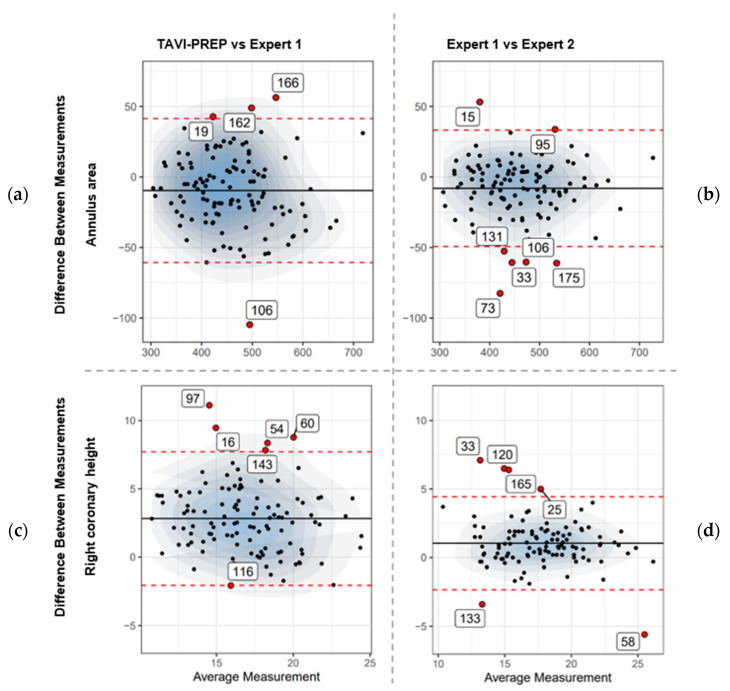
Bland–Altman plots comparing the automatic pipeline to expert manual measurements from 3Mensio. The black solid horizontal lines denote the mean, whereas the red dashed lines represent the 95% confidence interval boundaries. The first column, (**a**,**c**), depicts the measurements compared between the first expert and the TAVI-PREP algorithm, while the second column, (**b**,**d**), depicts the comparison between the two experts (*n* = 115 for area and *n* = 90 for RCH). (**a**,**b**) are the aortic annulus area while (**c**,**d**) are the right coronary height (RCH). Outliers are identified with their unique ID.

**Figure 5 diagnostics-13-03181-f005:**
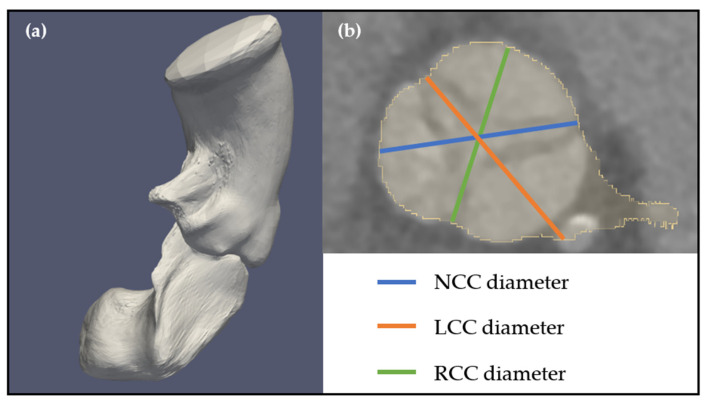
A scenario in which a segmentation failure disrupts the accuracy of the pipeline. In Panel (**a**), the mesh generated by the MeshDeformNet algorithm displays a significant segmentation error. Panel (**b**) presents a binary cross-section of the SOV plane, where measurements for NCC, RCC, and LCC are usually taken. Notably, the predicted LCC diameter is inaccurately estimated due to the incorrect segmentation. Since the error is not significant, the redundancy abilities of the algorithm are not triggered in this case.

**Figure 6 diagnostics-13-03181-f006:**
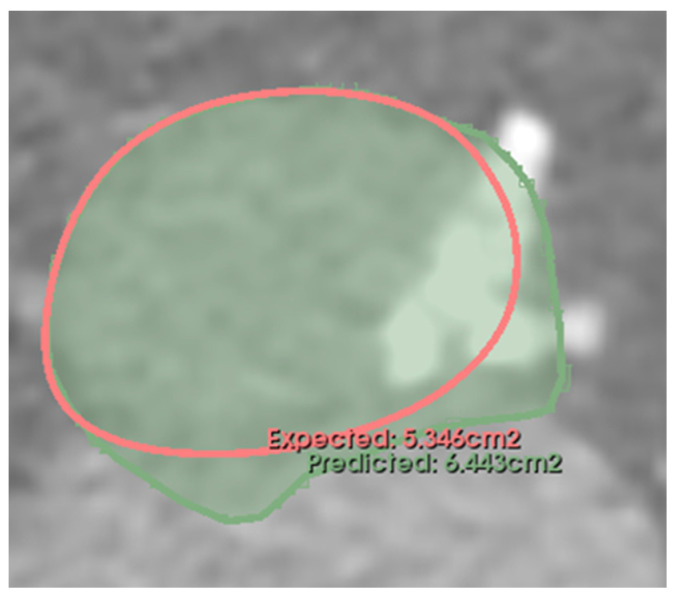
Case 106. The left ventricle is over-segmented, causing predicted measurements (area, perimeter, area-derived diameter, perimeter-derived diameter, minimum diameter, maximum diameter, and average diameter) to be overestimated. The scan has severe calcification at the LVOT and aortic annulus level (white region on the figure).

**Table 1 diagnostics-13-03181-t001:** Comparison of absolute error [95% CI] across the different anatomical structures between TAVI-PREP and previous works.

Measurements	Expert 1 vs. Expert 2 *n* = 115	TAVI-PREP vs. Expert 1 *n* = 200	Saitta et al. [21] *n* = 178	Astudillo et al. [22] *n* = 100	Elattar et al. [32] *n* = 40
Annulus area [mm^2^]	−8.08 [−49.30, 33.14]	**−9.65 [** **−60.65, 41.36** **]**	NA	NA	NA
Annulus perimeter [mm]	−0.83 [−4.56, 2.89]	**−0.72 [−5.35, 3.90]**	−1.8 [−8.06, 11.74]	NA	NA
Annulus area-derived diameter [mm]	−0.21 [−1.33, 0.91]	**−0.24 [−1.56, 1.09]**	**0.07 [−0.24, 0.38] ***	NA	NA
Annulus perimeter-derived diameter [mm]	−0.27 [−1.44, 0.91]	**−0.23 [−1.71, 1.25]**	NA	NA	NA
Annulus diameter minimum [mm]	−0.17 [−1.70, 1.35]	**−0.10 [−1.80, 1.59]**	0.89 [−2.8, 4.62]	NA	NA
Annulus diameter maximum [mm]	−0.24 [−2.06, 1.59]	**0.04 [−2.11, 2.20]**	0.51 [−2.79, 3.81]	NA	NA
Annulus diameter average [mm]	−0.20 [−1.54, 1.14]	**−0.03 [−1.58, 1.52]**	0.52 [−2.96, 4.00]	NA	0.48 [−2.26, 3.24]
SNTJ diameter average [mm]	0.79 [−1.52, 3.09]	**−0.33 [−1.98, 1.31]**	0.05 [−1.98, 2.07]	NA	NA
Left coronary height (LCH) [mm]	0.45 [−2.28, 3.17]	−0.05 [−4.00, 3.89]	NA	**0.54 [−2.46, 3.54]**	NA
Right coronary height (RCH) [mm]	0.45 [−2.35, 4.40]	2.82 [−2.06, 7.71]	NA	**−0.16 [−4.09, 3.78]**	NA

**Bold values highlight the method with an error closest to the interobserver variability**. NA (Not Available) values mean that the authors did not report this value in their analysis. * Potential error, see discussion below.

**Table 2 diagnostics-13-03181-t002:** Comparison of Pearson correlation coefficients [95% CI] across the different anatomical structures between TAVI-PREP and previous works, when available.

Measurements	Expert 1 vs. Expert 2 *n* = 115	TAVI-PREP vs. Expert 1 *n* = 200	Saitta et al. [21] *n* = 178	Astudillo et al. [22] *n* = 100	Elattar et al. [32] *n* = 40
Annulus diameter average	0.95 [0.93, 0.96]	**0.93 [0.91, 0.95]**	NA	NA	0.84
Left coronary height (LCH)	0.92 [0.89, 0.94]	**0.80 [0.74, 0.85]**	NA	0.80	NA
Right coronary height (RCH)	0.86 [0.82, 0.90]	0.72 [0.64, 0.78]	NA	**0.80**	NA
Average LCH-RCH	0.89 [0.85, 0.92]	0.76 [0.69, 0.82]	NA	**0.80**	0.73

**Bold values highlight the method with an error closest to the interobserver variability.** NA (Not Available) values mean that the authors did not report this value in their analysis.

## Data Availability

This work used the publicly available datasets MM-WHS available at: https://zmiclab.github.io/zxh/0/mmwhs/data.html accessed on 31 August 2023 [18,19,20]. The private dataset used to enhance the segmentation performance cannot be publicly shared. Contact M.S.-C. for study design questions and D.C. for technical questions.

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
