# Peer review of "TAVI-PREP: A Deep Learning-Based Tool for Automated Measurements Extraction in TAVI Planning"

_diagnostics, 2023, doi:10.3390/diagnostics13203181_

Round 1
Reviewer 1 Report
The study is good and interesting, and there are some points that should be addressed as follows:
1. Abstract is incomplete. The most important results of the study must be discussed.
2. The introduction is unsatisfactory. The authors started with a satisfactory and good start, but did not address the problem and its solution. The importance of artificial intelligence in solving the problem of manual diagnosis must be discussed.
3. Discussing the most important contributions at the end of the introduction and concluding the introduction with outlines for dividing the rest of the study.
4. Can you support the “2.2. Segmentation” section with pictures and the most important segmentation areas after the segmentation process?
5. Was the segmentation method performed before inputting the images into the deep learning models?
6. How was hyperparameter tuning set and what deep learning models were applied in the study.
7. The study must be supported by a figure representing the methodology to make it easy for readers to follow it.
8. Are there results such as accuracy, sensitivity, specificity, AUC, etc.?
9. What are the most important limitations of the study and what are future works?
10. The conclusions section must be repeated to suit the study and review the most important results of the study.
Minor editing of English language required
Reviewer 2 Report
Thank you very much for the opportunity to review this interesting publication.
Overall, the article is well written and contributes to the development of TAVI.
However, there are some doubts that need to be resolved.
To what extent are automatic measurements able to predict post-procedure complications, such as paravalvular leaks?
Round 2
Reviewer 1 Report
Accept in present form.
Minor editing of English language required.